# *Pseudomonas* and *Curtobacterium* Strains from Olive Rhizosphere Characterized and Evaluated for Plant Growth Promoting Traits

**DOI:** 10.3390/plants11172245

**Published:** 2022-08-29

**Authors:** Martino Schillaci, Aida Raio, Fabiano Sillo, Elisa Zampieri, Shahid Mahmood, Muzammil Anjum, Azeem Khalid, Mauro Centritto

**Affiliations:** 1National Research Council, Institute for Sustainable Plant Protection, Strada delle Cacce 73, 10135 Torino, Italy; 2National Research Council, Institute for Sustainable Plant Protection, Via Madonna del Piano 10, 50019 Sesto Fiorentino, Italy; 3Department of Environmental Sciences, PMAS Arid Agriculture University, Rawalpindi 46300, Pakistan

**Keywords:** olive rhizosphere, abiotic stress, taxonomy, plant growth promoting (PGP) bacteria, *Triticum durum*, *Curtobacterium*, *Pseudomonas*

## Abstract

Plant growth promoting (PGP) bacteria are known to enhance plant growth and protect them from environmental stresses through different pathways. The rhizosphere of perennial plants, including olive, may represent a relevant reservoir of PGP bacteria. Here, seven bacterial strains isolated from olive rhizosphere have been characterized taxonomically by 16S sequencing and biochemically, to evaluate their PGP potential. Most strains were identified as *Pseudomonas* or *Bacillus* spp., while the most promising ones belonged to genera *Pseudomonas* and *Curtobacterium*. Those strains have been tested for their capacity to grow under osmotic or salinity stress and to improve the germination and early development of *Triticum durum* subjected or not to those stresses. The selected strains had the ability to grow under severe stress, and a positive effect has been observed in non-stressed seedlings inoculated with one of the *Pseudomonas* strains, which showed promising characteristics that should be further evaluated. The biochemical and taxonomical characterization of bacterial strains isolated from different niches and the evaluation of their interaction with plants under varying conditions will help to increase our knowledge on PGP microorganisms and their use in agriculture.

## 1. Introduction

The rhizosphere is the layer of soil reached by root exudates and, due to the fact that those compounds can be used by microbiota as nutrients, it harbors a higher microbial population than soil not associated with plant roots [1]. As a result, the rhizosphere is an extremely dynamic environment where plants and microorganisms mutually influence each other [2]. Plant growth promoting (PGP) bacteria typically inhabit the rhizosphere of plants and promote plant growth through direct and indirect pathways [3]. The significance of PGP bacteria in plant nutrition and pathogen biocontrol is increasingly perceived, and in recent years numerous studies investigated their application for the improvement of crops’ performance in an environmentally and economically sustainable way [4,5]. 

Among all microbial niches, the rhizosphere of perennial plants is of particular interest as a source of PGP bacteria [6,7]. The long life cycle of perennial allows them to establish complex relationships with soil microbiota, which can help them to withstand year-round stress [8].

Olive (*Olea europaea*) is one of the most anciently domesticated crops, and traditionally requires low agricultural inputs [9]. It is likely that olive’s adaptability is also due to the interaction with soil beneficial microorganisms, and olive-associated bacteria have displayed several PGP traits both in vitro [10] and in vivo [11].

Being impractical to run in vivo experiments involving a wide number of bacterial isolates, the typical procedure to study rhizosphere bacteria is to primarily test them for beneficial traits and subsequently to inoculate plants with those strains that gave the best results [12,13]. Most studies isolate bacteria from the rhizosphere of plants subjected to a specific stress and test the ability of those bacteria to improve plant resistance to such stress [5]. Bacteria capable of growing in suboptimal environments have also been found in soils not affected by harsh conditions [14,15], and few studies have focused on the positive interaction between plants and bacteria isolated from the rhizosphere of such soils [16,17].

Plants and their beneficial microbiome often developed species-specific interactions by co-evolving in the course of time [18].

However, plants can also benefit from the interaction with bacteria isolated from the rhizosphere of other plant species [16,19,20,21]. The most commonly reported mechanisms of plant growth promotion are the improvement of plant nutritional status by P solubilization [16,20] and N-fixation [19] and the bacterial synthesis of PGP compounds, such as indole acetic acid (IAA) [19,20] and aminocyclopropane-1-carboxylic acid (ACC) deaminase [19,20,21].

Carefully identifying the bacteria isolated in a specific environment and linking them to their potential PGP traits is of great importance for the construction of a comprehensive database. In recent years, several studies have pursued this goal, providing taxonomic classifications of PGP bacteria interacting with various plants and describing methods for their identifications [22,23].

In this study, seven bacterial strains isolated from the olive rhizosphere were taxonomically characterized, and their PGP traits were assessed in vitro. Those with higher PGP potential were tested for their capacity to improve the germination and early development of *Triticum durum* subjected to osmotic or salinity stress.

## 2. Results

### 2.1. Strains Biochemical Characterization

Seven bacterial strains isolated from olive rhizosphere soil were characterized for various PGP traits, reported in Table 1. Isolated strains behaved differently with regard to their mobility, ability to produce specific biomolecules and to use tricalcium phosphate as P source. PK6, PK14 and PK18 were Gram negative. Strains PK5, PK6, PK14, PK18, and PK30 showed swimming and swarming ability, while strains PK11 and PK19 showed swarming ability only. Strains PK11, PK18, and PK19 were able to produce proteases, and PK5 and PK30 produced lipolytic enzymes. No strains produced exopolysaccharides (EPSs). Strains PK6, PK14, PK18, and PK30 were able to solubilize phosphate. Strains PK6, PK18 and PK19 produced IAA, and strains PK11, PK14, PK19, and PK30 produced N-acyl homoserine lactones (AHLs). None of the strains was able to grow using ACC as N source, and no gene encoding for ACC deaminase was found by tentative amplification using degenerative primers.

### 2.2. Molecular Characterization of Strains: 16S BLAST Results and Phylogenetic Analyses

The strain molecular characterization was performed sequencing the 16S rRNA gene. All PCR amplicons were successfully sequenced, providing sequences from 1303 to 1405 bp with the exception of PK5, for which an amplicon of 954 bp was obtained. The sequences were deposited in the GenBank database (for the accession numbers and full sequences, see Appendix A) and compared against the NCBI database for their characterization. PK5 had the highest similarity to *Bacillus subtilis* strain BD18-B23 [24], PK6 to *Pseudomonas* sp. PG-2010-28 [25], PK11 and PK19 to *Bacillus* sp. Iri 4/2 [26], PK14 to *Pseudomonas brassicacearum* subsp. *neoaurantiaca* strain CIP 109457 [27], PK18 to *Pseudomonas* sp. strain 3–8 [28] and PK30 to *Curtobacterium flaccumfaciens* strain C-S-R1-2 [29]. For the similarity scores of the five best hits of each PK strain against the NCBI database, see Appendix A.

Three strains were selected for deeper taxonomical characterization, as they gave the most promising results in the biochemical tests. Strains PK6 and PK18 were chosen as they both were able to synthesize IAA and had the highest P-solubilization capacity, while PK30 was chosen since it could solubilize P, had lipolytic capacity and produced AHLs (Table 1). The phylogenetic tree of PK6, PK18, and PK30 confirmed the results obtained from the BLAST analyses, placing PK6 closest to *Pseudomonas* sp. PG-2010-28, PK18 closest to *Pseudomonas* sp. strain 3–8, and PK30 closest to *Curtobacterium flaccumfaciens* strain C-S-R1-2 (Figure 1).

### 2.3. Bacterial and Triticum durum Growth under Osmotic or Salinity Stress

Strains PK6, PK18, and PK30 were tested for growth under osmotic or salinity stress. In the osmotic stress test, all strains were able to grow on nutrient broth with added glucose (NBG) [30] containing 20% polyethylene glycol (PEG) (Table 2), while in the salinity stress test PK6 was capable of growing on nutrient glucose agar (NGA) [30] containing 5% NaCl and PK18 and PK30 grew on NGA containing 7.5% NaCl (Table 2).

Strains PK6, PK18, and PK30 were then used as inocula on *Triticum durum* (*T. durum*) seeds germinating under osmotic or salinity stress. Results showed that none of the tested strains significantly improved the performance of *T. durum* under osmotic (PEG) or saline (NaCl) stress. Osmotic stress greatly affected the seedling vigour index (SVI) [31] of non-inoculated plants, which decreased by 78.7% at 16% PEG and by 93.3% at 24% PEG. At 30% PEG, no seed germination was observed (Figure 2a). Similarly, non-inoculated *T. durum* was highly susceptible to salinity stress, which decreased SVI by 53.2% at 0.5% NaCl, 78.1% at 1% NaCl, and by 99.8% at 2% NaCl (Figure 2b). Six days after the inoculation, the SVI of PK18-inoculated plants was 97.8% higher (*p*-value = 0.009) than that of non-inoculated plants when germinated in the absence of PEG or NaCl (Figure 2). The DNA extraction, PCR with specific primers, amplicon visualization and sequencing confirmed that the PK18-inoculated samples contained PK18 DNA, while the non-inoculated samples did not contain DNA from either PK6, PK18 or PK30.

All other strains did not significantly affect *T. durum*’s development in the same conditions, and no increase in SVI was observed for any bacterial treatment performed at higher PEG or NaCl concentrations (Figure 2).

## 3. Discussion

In this study, a biochemical characterization of seven bacterial strains isolated from the olive rhizosphere was carried out to assess their in vitro PGP potential (Table 1). Among the studied strains, PK6, PK18, and PK30 were further characterized as they showed the most promising traits. All three strains were capable of swimming and swarming, which allow chemotaxis and colonization of plant tissues [20]. Similarly, all three strains were capable of P solubilization, a desirable trait for PGP bacteria particularly in P-poor environments, as plants can adsorb the mobilised P and use it for their growth [32]. PK30 could produce AHLs, quorum sensing-associated biomolecules that can also play a stimulatory role on plant tissues development [33]. Finally, PK6 and PK18 could synthesize IAA, a well-known plant growth promoter, and one of the most sought-after traits in potential PGP bacteria, as it has been reported to increase plant growth by enhancing cell elongation [6].

Genera *Bacillus* and *Pseudomonas* were the most common among those considered in this study, and bacteria belonging to these genera were also found by Marasco et al. [10] among the microbiota associated with olive roots. In their tests, similarly to ours, *Pseudomonas* strains performed generally better than *Bacillus* in regard to their P-solubilization and IAA production capacity (Table 1).

Phylogenetic analysis based on 16S rRNA gene sequences of PK6, PK18 and PK30 strains allowed to compare them to the genetically close strains present in literature (Figure 1). PK6 was closest to *Pseudomonas* sp. PG-2010-28 but, to our knowledge, no information is available on the interactions between this strain and plants. Looking at the closely related species, *P. fuscovaginae* is a well-known pathogen of various cereals [34], while *P. alcaligenes* is a strain with promising PGP traits, whose volatile organic compounds have been successfully used against drought stress [35]. Strain PK18 was genetically closest to *Pseudomonas* sp. strain 3–8, and also in this case no effects of the interaction between this strain and plants are reported. Among similar species, *P. gessardii* was recently reported to improve sunflower’s response to Pb-toxicity [36]. PK30 had the highest similarity to *Curtobacterium flaccumfaciens* strain C-S-R1-2, a diazotroph strain associated with the halotolerant plant *Salicornia europaea* [29]. *C. flaccumfaciens* is mostly known as pathogenic species, but it has also shown PGP traits, including the improvement of drought-tolerance in barley [37].

Overall, the results of the phylogenetic analysis showed how closely related bacterial strains can behave differently from each other, and no generalization on their behavior in association with plants can be made. The interactions between plants and bacteria are often unpredictable: the same bacterial strain can be beneficial to a crop and detrimental to another [38], or beneficial only at certain growing conditions [39,40]. For these reasons, it is still extremely important to test each strain, both in vitro and in vivo, to evaluate it axenically but even more importantly in its interaction with specific crops and at specific conditions.

Bacterial strains PK6, PK18, and PK30 demonstrated good resistance to osmotic and salinity stress. In the osmotic stress test, all strains were able to grow on NBG containing 20% PEG, while in the saline stress test PK6 was capable of growing on NGA containing 5% NaCl and PK18 and PK30 grew on NGA containing 7.5% NaCl. Osmotolerant and halotolerant bacteria have been previously isolated from normal (i.e., non-extreme) environments: Echigo et al. [14] found strains capable of growing in media with 20% NaCl or higher in various substrates not characterized by high salinity. Halotolerant bacteria have developed various mechanisms to cope with osmotic stress, such as the synthesis of hydrolytic enzymes [41], ion pumps for the extrusion of Na^+^ from the cell and production of compatible solutes to increase the cell osmotic potential [42]. PK18 and PK30 displayed respectively proteolytic and lipolytic capacity, while PK6 lacked both (Table 1). This might partially explain why PK18 and PK30 performed better than PK6 on saline media.

Results obtained from the germination experiment showed that none of the tested strains significantly improved the performance of *T. durum* under osmotic or saline stress, despite showing good in vitro potential: besides growing on substrates supplemented with PEG or NaCl, PK18 and PK30 produced hydrolytic enzymes and PK6 and PK18 tested positive for IAA production (Table 1). IAA, which belongs to the auxins class, is normally produced by plants, and it is involved in major processes as vascular differentiation, cell elongation and apical dominance but also in the plant response to various stresses, including salinity and osmotic stress [43,44]. Zörb et al. [45] compared the auxin levels of two maize cultivars with different resistance to saline stress and report that high salinity decreased IAA level in the more susceptible variety, while IAA was unaffected in the resistant one. PGP bacteria could be involved in the modulation of IAA in stressed plants: wheat plants subjected to drought and saline stress had increased endogenous IAA levels and improved growth when inoculated with various bacterial strains [46]. In our study, IAA might have played a decisive role in improving the performance of PK18-inoculated plants in the absence of osmotic and salinity stress. At 0% PEG/NaCl, inoculated plants had almost double SVI than their non-inoculated counterparts, due mainly to +63.4% longer seedlings (data not shown). IAA-producing PGP bacteria have been reported to improve the germination and early development of various crops [47,48]. Considering the conditions of the experiment, it is unlikely that other PGP traits found in PK18, such as P solubilization or proteolytic activity, may have played a significant role in the promotion of plant growth. It is instead possible that other mechanisms, not measured in our in vitro tests, were responsible for the observed results. Bacterial metabolism often changes greatly during the interaction with plants, developing new pathways that cannot be observed in axenic cultures [37].

## 4. Materials and Methods

### 4.1. Strains Biochemical Characterization

Bacterial strains PK5, PK6, PK11, PK14, PK18, PK19 and PK30 were isolated in Italy from the rhizosphere of olive trees (cultivars Leccio del Corno, Frantoio and Moraiolo) located in a non-irrigated area at San Casciano in Val di Pesa (Florence, Italy). In order to increase the sources of potential PGP bacteria, plants of different ages were selected for the collection of the rhizosphere soil. About 30 cm hole was dug around the roots of olive plants. Some parts of lateral roots were harvested for the collection of rhizosphere soil. The soil along with root and root hairs were stored in polythene bags. Loosely attached soil was removed by gentle shaking, while rhizosphere soil was removed by dipping the multiple roots in 250 mL sterile water to form composite soil suspension. samples. Dilution plate technique was used for the bacteria isolation. A series of dilutions were prepared from the suspension of rhizosphere soil, and 200 µL from each dilution were plated onto nutrient glucose agar (NGA) spiked with 0.5% tricalcium phosphate. Plates were incubated at 27 °C for 48 h, after which seven bacterial strains were isolated. Pure isolates were stored in cryovials containing a sterile 30% glycerol solution at −80 °C for long term storage.

The following biochemical tests were performed by using appropriate reference bacterial species. All tests were performed in triplicate.

Bacterial motility

Single colonies of each strain were picked-up with a sterile needle and point inoculated in the middle of Petri dishes containing tryptone swim agar (1% tryptone, 0.5% NaCl, 0.3% agar) or swarm agar (0.5% Bacto Agar, 8 g of nutrient broth, 5 g of dextrose). The radial growth of the colonies was measured after 48 h of incubation at 27 °C [49].

Proteolytic activity

Bacterial suspensions of 1 × 10^7^ colony forming units (CFUs)/mL were prepared by measuring the OD_530_. Twenty µL of each suspension were spot inoculated in the middle of Petri dishes containing Luria Bertani (LB) medium with 1.5% *w*/*v* agar and supplemented with 2% skim milk. Plates were subsequently incubated at 27 °C for 48 h. The formation of a clarification halo around the macro colonies developing in the middle of the Petri dishes indicated a positive result [50].

Lipolytic activity

Cell suspensions were prepared as previously described. Then 20 µL of each suspension were spot inoculated in the middle of Petri dishes containing LB agar amended with 1% Tween 40. The formation of a halo around the macro colonies developing in the middle of the Petri dishes indicated a positive result [51].

EPSs production

Exopolysaccharides (EPSs) production was evaluated on yeast extract mannitol medium (YEM; 0.5 g of yeast extract, 4 g of mannitol, 15 g of agar). Strains were streaked on YEM and on LB (negative control) agar plates to obtain isolated colonies, which were incubated for 48 h at 27 °C. After incubation, the morphology of colonies on YEM and LB agar was compared [52].

Phosphate solubilization activity

Four strains per plate were stabbed on Pikovskaya (PVK) agar medium using sterile toothpicks. The halos developing around colonies were measured after 14 days of incubation at 27 °C [53].

IAA production

Bacterial strains were inoculated with toothpicks into a grid pattern on the surface of Petri dishes containing LB agar amended with 5 mM L-tryptophan (LBT) or unamended LB agar. Each inoculated plate was then overlaid with a sterile disk of Whatman paper n.2 and incubated at 27 °C until the colonies reached 0.5–2 mm diameter. The paper disks were then removed and treated with Salkowsky reagent. Bacteria producing indole acetic acid (IAA) were identified by the formation of a characteristic red halo within the paper disk immediately surrounding the bacterial colony [54].

AHLs production

N-acyl homoserine lactones (AHLs) production was assessed by T-streak technique, using the reporter strain NT1 (pZLR4) of *Agrobacterium tumefaciens* after a period of incubation of 24–48 h [55].

ACC deaminase

Aminocyclopropane-1-carboxylic acid (ACC) deaminase synthesis was tested at the molecular level by tentatively amplifying the ACC deaminase-encoding gene *acdS* using four combinations of degenerative primers [56] on DNA extracted from the strains (see Section 4.2). To further validate the obtained results, the strains were tested in vitro for the ability to use ACC as sole nitrogen source, following the protocol described by Penrose and Glick [57] with some modifications. Briefly, plates containing solid DF salts minimal medium were spread with ACC (30 µL plate^−1^) just prior to use and let dry under a laminar hood. Three spots (20 µL) of each bacterial culture grown on KB broth for 48 h at 27 °C were placed on the surface of the agar plates. Plates were incubated at 27 °C for three days and then bacterial growth was checked. DF agar plates without ACC and DF agar plates amended with (NH_4_)_2_SO_4_ were used as negative and positive control respectively.

Gram reaction

The rapid non staining method described by Buck [58] was used to determine the Gram reaction of the bacterial isolates.

### 4.2. Molecular Taxonomic Identification of the Selected Strains

Strains were grown overnight in nutrient broth with added glucose (NBG) on a shaking incubator (28 °C, 180 rpm), then DNA was extracted using the E.Z.N.A.^®^ Bacterial DNA Kit (Omega Bio-tek, Norcross, GA, USA).

16S rRNA gene sequence of bacterial strains was amplified with primers 27F (5′ GAGAGTTTGATCCTGGCTCAG 3′) [59] and 1495R (5′ CTACGGCTACCTTGTTACGA 3′) [60] using the Hot Start PCR Master Mix (Thermo Fisher Scientific, Waltham, MA, USA). PCR settings for 27-1495 amplification were the following: 2 min at 95 °C; 30 s at 94 °C, 30 s at 55 °C, 90 s at 72 °C (34 cycles); 7 min at 72 °C.

The obtained amplicons were purified using the QIAquick^®^ PCR Purification Kit (Qiagen, Hilden, Germany), sequenced at Ludwig Maximilian University Sequencing Service (Martinsried, Germany), and compared with the National Center for Biotechnology Information (NCBI) sequence database using the basic local alignment search tool (BLAST) tool [61]. Using the software MEGA X [62], phylogenetic trees were then constructed using the 27-1495 sequences of the strains that performed better in the biochemical tests (see Section 2.1). The settings for the phylogenetic tree were the following: statistical method, maximum likelihood; test of phylogeny, bootstrap method with 100 re-samplings of the sequence alignment.

### 4.3. Bacterial Growth Assay under Osmotic or Saline Conditions

Three strains were selected for their ability to produce IAA (PK6 and PK18) and AHLs (PK30), solubilize P (PK6, PK18 and PK30), and lipolytic activity (PK30). Strains were grown on nutrient media with varying PEG or NaCl content.

To test the bacterial growth under osmotic stress, NBG medium containing 0% (control), 10%, 20%, 30% or 40% PEG 6000 [63] was inoculated with the selected bacterial strains and placed in a shaking incubator set at 26 °C and 180 rpm. After 24 h, bacterial growth was verified by spreading aliquots from each inoculated medium on NGA plates, which were then stored in the dark at 26 °C, and bacterial growth was verified after 72 h.

To test the bacterial growth under saline stress, bacterial colonies were streaked on NGA plates containing 0.5% (control), 2.5%, 5%, 7.5% or 10% NaCl [64]. Again, inoculated plates were then stored in the dark at 26 °C, and bacterial growth was verified after 72 h.

### 4.4. Assessment of Germination under Osmotic or Saline Stress of T. durum Seeds Inoculated with PGP Bacteria

*T. durum* seeds cv. Marco Aurelio (Società Italiana Sementi, San Lazzaro di Savena, Italy) were sterilized by immersion in 70% *v*/*v* EtOH for 60 s and in 2.5% *v*/*v* NaClO for 30 min, then rinsed five times with sterile distilled water (SDW) and stored overnight in the dark at 4 °C in SDW.

The three strains previously tested (see Section 4.3) were used in this experiment. The inoculum was prepared by growing bacteria in NBG in a shaking incubator set at 26 °C and 180 rpm. When the PK6 and PK18 cultures reached respectively the OD_600_ of 1.6 and 1.5, equivalent to approximately 10^12^ CFUs/mL, they were diluted with sterile 0.9% *w*/*v* NaCl solution to reach the inoculum concentration of 10^7^ CFUs/mL. Seeds were inoculated by transferring them into the bacterial solution for 120 min in the shaking incubator set as previously described. Non-inoculated seeds were transferred into the 0.9% *w*/*v* NaCl solution, all other conditions kept unchanged. After the inoculation, seeds were placed in Petri dishes (30 seeds per treatment) containing different PEG 6000 (16.1, 24.1, and 30.2% *w*/*v*) [65] or NaCl (0, 0.5, 1, and 2% *w*/*v*) [66] concentrations and stored in the dark for 6 days. Finally, germinated seeds were counted, seedlings were collected, and root and shoot length were measured with a ruler to determine the SVI, a parameter that takes in account both the germination rate of seeds and the growth of germinated seedlings [31].
SIV = (root length + shoot length) × germination rate 

### 4.5. Validation of Colonization by the Inoculated Bacteria

At the end of the experiment, three plant samples were randomly chosen to confirm the bacterial inoculation in the treatments which differed significantly (see Section 2.3). The presence of the inoculated strains was verified by DNA extraction using the DNeasy Mini Kit (Qiagen, Hilden, Germany), and the DNA was used as template for a PCR using the *Pseudomonas*-specific primers PK18F (5′ GTGGGGTAATGGCTCACCAA 3′) and PK18R (5′ AAGAGCTCAAGGCTCCCAAC 3′), designed on PK18′s 16S rRNA gene sequence (see Appendix A) with the NCBI primers designing tool [67]. Primers were synthesized by Eurofins Genomics GmbH (Ebersberg, Germany). The obtained amplicons were visualised on 1% agarose gel and sequenced by Eurofins Genomics GmbH.

### 4.6. Statistical Analyses

After verification of normality of the data distribution with a Shapiro–Wilk test and homogeneity of variance assessed by a Levene’s test, the significance of the difference among parameters of treatments was determined by using one-way ANOVA followed by a Tukey HSD test. We consider as significantly different only those parameters that had a significance level (*p*-value) < 0.05.

## 5. Conclusions

In this study, we tested the PGP potential of seven strains isolated from the olive rhizosphere, in vivo and when interacting with *T. durum* seeds subjected to osmotic or saline stress. The different bacteria failed to improve the SVI of plants in a saline environment but showed several promising PGP traits when grown in vitro. The potential of many of those strains as beneficial bacteria had not been evaluated before, and we believe this study provided some insights on rhizosphere bacteria and their use in agriculture, by characterizing them both biochemically and taxonomically. Defining the taxonomy of the isolated bacterial species and linking them to their PGP traits, which are often specific to the growing conditions and the interacting plant species, can provide valuable information on the use of beneficial bacteria to improve plant performance.

## Figures and Tables

**Figure 1 plants-11-02245-f001:**
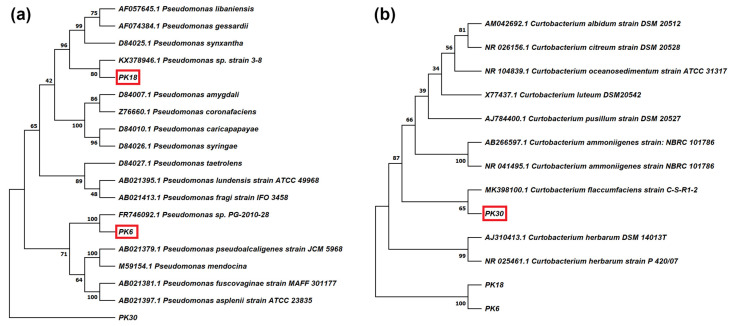
Phylogenetic tree based on 16S rRNA gene sequences of (**a**) PK6 and PK18 strains identified as *Pseudomonas* spp. and (**b**) PK30, identified as *Curtobacterium* sp. PK30 was used as outgroup for building the *Pseudomonas* tree, and PK6 and PK18 were used as outgroup for building the *Curtobacterium* tree.

**Figure 2 plants-11-02245-f002:**
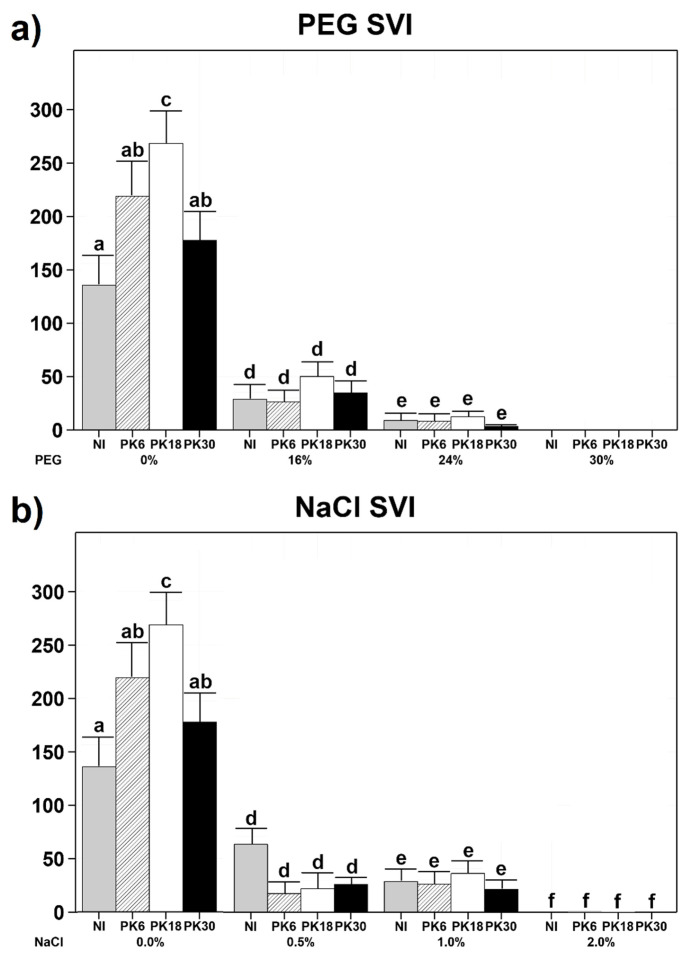
Barplots showing seedling vigor index (SVI) of *Triticum durum* inoculated or non-inoculated with bacterial strains isolated from olive rhizosphere and germinated for 6 days in the dark under osmotic or salinity stress. NI = non-inoculated. (**a**) Plants germinated at varying PEG concentrations. At 0% PEG, *n* = 19 non-inoculated, *n* = 21 PK6-inoculated, *n* = 23 PK18-inoculated and *n* = 24 PK30-inoculated. At 16% PEG, *n* = 6 non-inoculated, *n* = 7 PK6-inoculated, *n* = 8 PK18-inoculated and *n* = 6 PK30-inoculated. At 24% PEG, *n* = 5 non-inoculated, *n* = 5 PK6-inoculated, *n* = 6 PK18-inoculated and *n* = 6 PK30-inoculated. At 30% PEG, no germination was observed in any of the treatments. (**b**) Plants germinated at varying NaCl concentrations. At 0% NaCl, *n* = 19 non-inoculated, *n* = 21 PK6-inoculated, *n* = 23 PK18-inoculated and *n* = 24 PK30-inoculated. At 0.5% NaCl, *n* = 9 non-inoculated, *n* = 4 PK6-inoculated, *n* = 4 PK18-inoculated and *n* = 6 PK30-inoculated. At 1% NaCl, *n* = 15 non-inoculated, *n* = 14 PK6-inoculated, *n* = 15 PK18-inoculated and *n* = 15 PK30-inoculated. At 2% NaCl, *n* = 6 non-inoculated, *n* = 9 PK6-inoculated, *n* = 10 PK18-inoculated and *n* = 6 PK30-inoculated. Means + standard errors are presented. Different letters above bars represent significant differences (*p* < 0.05) among treatments according to Tukey’s HSD outcomes.

**Table 1 plants-11-02245-t001:** Results obtained by biochemical and physiological tests on bacterial strains isolated from olive rhizosphere. Swimming, swarming and P solubilization results are displayed as colony diameter. EPSs = exopolysaccharides, IAA = indole acetic acid, AHLs = N-acyl homoserine lactones, ACCD = aminocyclopropane-1-carboxylic acid (ACC) deaminase. Data represents the average of three replicates ± standard deviation.

Strain	Gram Reaction	Swimming (mm)	Swarming (mm)	Proteo-lysis	Lipolysis	EPSs	P Solubilization (mm)	IAA Production	AHLs Production	ACCD Production
PK5	+	40.3 ± 1.53	19.3 ± 1.15	−	+	−	−	−	−	−
PK6	−	19 ± 0	3.7 ± 0.58	−	−	−	3 ± 0	+	−	−
PK11	+	2 ± 0	4.7 ± 1.15	+	−	−	−	−	+	−
PK14	−	10.3 ± 0.58	4.3 ± 0.58	−	−	−	2 ± 0	−	+	−
PK18	−	17.3 ± 0.58	5 ± 1	+	−	−	3.7 ± 0.58	+	−	−
PK19	+	2 ± 0	14 ± 0	+	−	−	−	+	+	−
PK30	+	12.3 ± 0.58	4.3 ± 0.58	−	+	−	1.3 ± 0.58	−	+	−

**Table 2 plants-11-02245-t002:** Growth of bacterial strains on NGA with different NaCl levels. y = growth detected, n = growth not detected.

	NaCl Content					PEG Content				
Strain	0.5%	2.5%	5%	7.5%	10%	0%	10%	20%	30%	40%
PK6	y	y	y	n	n	y	y	y	n	n
PK18	y	y	y	y	n	y	y	y	n	n
PK30	y	y	y	y	n	y	y	y	n	n

## Data Availability

The data presented in this study are available in the Appendix A.

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
