# Peer review of "Pseudomonas and Curtobacterium Strains from Olive Rhizosphere Characterized and Evaluated for Plant Growth Promoting Traits"

_plants, 2022, doi:10.3390/plants11172245_

Round 1

Reviewer 1 Report

This manuscript describes and characterized several bacterial strains isolated form the rhizobacteria of olive trees. The isolated stains were tested for PGP characteristics, founding that three, out of the seven characterized strains, had traits that could let them to be considered growth promotors. Hence, these three strains were applied to seeds of Trticum to test their germination triggering potential. Results did not prove any of the strains as particularly interesting as PGP.

Although the study is well designed, there is a strong flaw in it that is the extremely low sample size in both the characterization of strains and the number or replicates used to characterize the isolated strains. The same can be said in regards the germination test. With such a low sample size it is not possible to believe that the results are representative of what happens in the rhizosphere of the olive trees. Due to the fact that the presented analyses are neither complicated, nor expensive, I would encourage the researchers to increase the results,  to develop more elaborated statistical analyses and to broaden the analyses to include more test to characterize the strains.

I have annotated several comments on the manuscript, that I’m including with this and that I believe that will help to improve the manuscript.

Author Response

Reviewer 1

This manuscript describes and characterized several bacterial strains isolated form the rhizobacteria of olive trees. The isolated stains were tested for PGP characteristics, founding that three, out of the seven characterized strains, had traits that could let them to be considered growth promotors. Hence, these three strains were applied to seeds of Trticum to test their germination triggering potential. Results did not prove any of the strains as particularly interesting as PGP.

Although the study is well designed, there is a strong flaw in it that is the extremely low sample size in both the characterization of strains and the number or replicates used to characterize the isolated strains. The same can be said in regards the germination test. With such a low sample size it is not possible to believe that the results are representative of what happens in the rhizosphere of the olive trees. Due to the fact that the presented analyses are neither complicated, nor expensive, I would encourage the researchers to increase the results, to develop more elaborated statistical analyses and to broaden the analyses to include more test to characterize the strains.

I have annotated several comments on the manuscript, that I’m including with this and that I believe that will help to improve the manuscript.

A: We thank the Reviewer for the comments. We did our best to address the Reviewer concerns (see below), to clarify unclear sentences and to better explain the performed analysis. To support or results, we performed additional statistical analyses as suggested (all info are now included in the revised version).

  • The sentence at line 40-42 is unclear

A: We have rephrased the sentence, to make it clearer.

  • The paragraph at line 56-68 is too long

A: We shortened this section, adding some PGP mechanisms, as suggested by the reviewer

  • Three replicates not being a sufficient sample size for the biochemical characterization of the isolated strains (line 96)

A: We found that most of the studies that perform in vitro tests to characterize newly isolated strains were based on the results of three replicates for each test (limited to our study, we report as examples Cassán et al., 2009; Mortuza et al., 2020; Rohban et al., 2009). The main aim to carry on these tests is to verify the presence of specific traits in each bacterial isolate and then to perform a preliminary selection among the isolated bacteria. In the light of these considerations we thought that a higher number of replicates was not necessary for the purpose of the study.

  • The reasons why strains PK6, PK18 and PK30 were selected after the biochemical tests is not clear (line 120, line 320)

A: We clarified the reasons for choosing those strains in lines 127-131 and 360-362.

  • Possibility that the DNA extracted from inoculated and non-inoculated plants came from a contamination (lines 138-140)

A: After seeds inoculation and assessment of phenotypic differences in seedlings, we wanted to verify the presence of the inoculated bacteria. In order to confirm the inoculation, we used a molecular approach based on PCR with specific primers that we designed to amplify the DNA of Pseudomonas spp., since the strain causing significant differences in SVI (PK18) was identified as belonging to such genus. We sequenced the amplicons that we obtained from the PCR, which allowed us to confirm that the amplicons obtained from PK18 inoculated plants matched exactly with the 16s of PK18. In non-inoculated seedlings, after PCR, a faint signal in agarose gel (Figure S1) was observed, and the corresponding amplicon was sequenced. The sequencing showed that those DNA fragments belonged to a different strain of Pseudomonas spp., not matching with PK18. Since we sterilized the seeds at the beginning of the experiment, we hypothesize that those Pseudomonas strains in non-inoculated plants were seed endophytes, which could not be killed by the sterilization step.

  • Simplify Figure 2 by removing unnecessary lines

A: We made an effort to improve Figure 2. In this version we used letters instead of lines and asterisk to inform about the differences among different treatments. We also removed the figure external borders

  • This is quite vague. Please, provide coordinates or a better site description, so as to be easily found.

A: The sampling site is located at San Casciano in Val di Pesa municipality (Florence, Italy). Coordinates of San Casciano in Val di Pesa municipality are 43°39′25″N 11°11′09″E.

  • As it is written, one gets the impression that all collected soil samples, regardless the parent plant, plant age or whether it was soil or rhizospheric soil, were pulled together to form a compose sample. This part needs to be developed and properly explained. Among other aspects, it is needed to explain why plants of different age were used, instead of having used plants of homogeneous age. It is likely that plants of different age exudate different compounds, hence attracting bacteria different from those from older or younger plants. Similarly, how was soil collected? At what depth? how much? how may samples formed a composed sample? (lines 236-242)

A: The aim of the isolation was to collect potential beneficial bacteria from the olive tree rhizosphere, keeping the source substrate as broad as possible. We were not interested in comparing the rhizosphere of different individuals or from plants of different ages (this is the same reason why we collected the rhizosphere of different cultivars). However, we agree that some details on soil sampling were missing. In this revised version, we expanded and reworded this section to make it clearer for the reader, by also adding requested info. Lines 274-279 have been amended for clarification related to composite sample. Similarly, sample collection process has been also further clarified.

  • Use full name the first time an acronym is used (materials and methods section)

A: We added the requested info.

  • All used techniques need a reference (materials and methods section)

A: The references were provided for each point of the bacterial characterization (references 49-58). For sake of clarity we moved the related reference at the end of each paragraph.

  • What are the bootstrap method replications? Where do they come from? (line 318)

A: We clarified this point in the revised version. The bootstrap method is one of the most commonly used test for assessing the reliability of inferred phylogenetic tree. This is performed bt MEGA software.

  • What is the rationale for the NaCl concentration in the agar medium (line 328)?

A: The 0.5% concentration is the standard NaCl concentration in the NGA medium. The other concentrations are commonly used in studies on bacterial resistance to salt stress. We added a reference on the used protocol, and we did the same for the PEG level of the same test. Furthermore, we added references for the chosen NaCl and PEG levels in the wheat germination test.

  • Three plants not being enough to verify the inoculation by PK18 (line 350)

A: We randomly selected 3 seedlings out of 30 used in the assay in order to make sure that the phenotype results we observed in the phenotype of inoculated plants was due to the presence of the inoculated bacteria, and that no cross contamination occurred in our non-inoculated plants. Considering the significant difference detected in SVI between non-inoculated and inoculated seedlings with PK18, we have considered the number of replicates sufficient for our purpose. For the quantitative analysis (SVI index) we used a higher number of plants per treatment (30). We have reworded the paragraph, to make it clearer.

  • Why did we use different primers for the 16s amplification and to verify the inoculation? (lines 353-354)

A: For the molecular characterization and subsequent phylogeny of the strains based on 16s amplification we used universal primers (27F and 1495R). Instead, for verifying the bacterial inoculation we used primers specifically designed on the inoculated strain, to increase specificity and reduce the possible amplification of DNA from the bacterial seed endophyte population, which would have created further bias in our analysis. For this reason, the primers for verification of inoculation were specifically designed on PK18 16s sequence.

Reviewer 2 Report

The manuscript submitted to me for revision examines the characterization of isolates from olive rhizosphere concerning their plant growth-promoting traits.  It possesses the structure recommended by the journal.

The abstract is well structured, and the keywords fit the manuscript’s content.

The introduction covers all experimental parts of the manuscript, and the literature reference is well selected and sufficiently detailed.

The results describe the work done in the study. Biochemical characterization consists in testing seven bacterial strains to several parameters: Gram staining, swimming, swarming, P solubilization, proteolysis, lipolysis, exopolysaccharides, indole acetic acid and N-acyl homoserine lactones production, the position of aminocyclopropane-1-carboxylic acid deaminase.

Furthermore, molecular characterization of the strains was done. In continuation, the ability of PK6, PK18 and PK30 strains to grow in nutrient broth under osmotic or salinity stress were tested. Finally, the authors applied an in-vitro test for stimulation of growth of seeds of Triticum durum, named Seedling vigour index.

The Materials and methods section explains the methodology clearly, which permits repetition.

Statistical results were analysed using Student’s t-test with a significance of p<0,05.

The conclusions are supported by the results and the discussion.

Author Response

Reviewer 2

The manuscript submitted to me for revision examines the characterization of isolates from olive rhizosphere concerning their plant growth-promoting traits. It possesses the structure recommended by the journal.

The abstract is well structured, and the keywords fit the manuscript’s content.

The introduction covers all experimental parts of the manuscript, and the literature reference is well selected and sufficiently detailed.

The results describe the work done in the study. Biochemical characterization consists in testing seven bacterial strains to several parameters: Gram staining, swimming, swarming, P solubilization, proteolysis, lipolysis, exopolysaccharides, indole acetic acid and N-acyl homoserine lactones production, the position of aminocyclopropane-1-carboxylic acid deaminase.

Furthermore, molecular characterization of the strains was done. In continuation, the ability of PK6, PK18 and PK30 strains to grow in nutrient broth under osmotic or salinity stress were tested. Finally, the authors applied an in-vitro test for stimulation of growth of seeds of Triticum durum, named Seedling vigour index.

The Materials and methods section explains the methodology clearly, which permits repetition.

Statistical results were analysed using Student’s t-test with a significance of p<0,05.

The conclusions are supported by the results and the discussion.

A: We thank the Reviewer for the positive comments. As suggested by Reviewer 1, we highlight that a new statistical analysis has been performed to support our data (ANOVA + Tukey HSD, significance of p<0.05).
